# Cone Beam Computerized Tomographic Analysis of Collum Angle of the Maxillary Central Incisors in Different Types of Malocclusion: Comparative Assessment in Saudi, Jordan and Egypt Subpopulation

**DOI:** 10.3390/diagnostics12061506

**Published:** 2022-06-20

**Authors:** Rakhi Issrani, Namdeo Prabhu, Sunitha Siddanna, Sarah Hatab Alanazi, Mohammad Khursheed Alam, Manay Srinivas Munisekhar, May Othman Hamza, Reham Fawzi Dawood Shabanah, Rasha Saleh Ali Gadh

**Affiliations:** 1Department of Preventive Dentistry, College of Dentistry, Jouf University, Sakaka 72388, Saudi Arabia; sara.alanazi@jodent.org (S.H.A.); mkalam@ju.edu.sa (M.K.A.); dr.srinivas.manay@jodent.org (M.S.M.); 2Department of Oral & Maxillofacial Surgery and Diagnostic Sciences, College of Dentistry, Jouf University, Sakaka 72388, Saudi Arabia; dr.namdeo.prabhu@jodent.org; 3Department of Public Health Dentistry, JSS Dental College and Hospital, Mysore 570015, India; drsunithas@gmail.com; 4Center of Transdisciplinary Research (CFTR), Saveetha Dental College, Saveetha Institute of Medical and Technical Sciences, Saveetha University, Chennai 602105, India; 5Department of Public Health, Faculty of Allied Health Sciences, Daffodil International University, Dhaka 1341, Bangladesh; 6Department of Prosthetic Dental Sciences, College of Dentistry, Jouf University, Sakaka 72388, Saudi Arabia; dr.may.hamza@jodent.org; 7General Dentist, South Faysaliah PHC, Arar 73211, Saudi Arabia; dr.reham.sh@gmail.com; 8Independent Researcher, Arar 73211, Saudi Arabia; dr.rasha.g@gmail.com

**Keywords:** 3D CBCT, collum angle, crown-root angle, malocclusion

## Abstract

Background: The collum angle (CA) is an extremely significant for patients who are undergoing orthodontic, dental implant restoration, prosthodontic and periodontic treatments. Aim and Objectives: To determine and compare the mean CA for maxillary central incisor in different types of malocclusion utilizing 3D Cone Beam Computerized Tomography (CBCT) images. The additional objectives were to determine and compare the mean CA for maxillary central incisor based upon the demographic characteristics among Saudi, Jordan and Egypt subpopulation and to test for significant differences in the CA of maxillary central incisor with different molar malocclusions. Methodology: A total of 400 CBCT images were included from the radiology archive at the College of Dentistry, Jouf University (Sakaka, Saudi Arabia). The CBCT images were divided into four groups based upon molar classifications. The selected records were used for the measurement of CA of maxillary central incisor using the measurement tool built into 3D:OnDemand software. Statistical analysis was done using independent t test and ANOVA to examine the differences between gender and races. Results: The mean CA for Class II div 2 exhibited significantly higher crown-root variation as compared other groups (*p* < 0.0001). Males sample showed greater value of CA for each group as compared to the females and this difference was statistically significant for all the groups other than for Class I (*p* < 0.05). The post hoc pairwise comparisons between the races showed statistically insignificant findings (*p* > 0.05). Significant difference was found on pairwise comparisons among different malocclusion groups other than for group Class I/Class II div 1 (*p* < 0.05). Conclusion: The CA of Class II div 2 group was the greatest as compared to other malocclusion groups. Males sample showed greater value of CA for each group as compared to the females and this difference was statistically significant for all the groups other than for Class I. Statistically insignificant difference was noted for the mean CA among different races whereas significant difference was found on pairwise comparisons among different malocclusion groups other than for group Class I/Class II div 1.

## 1. Introduction

Variability in tooth morphology inherently affects occlusion and its corresponding 3D positions [1,2]. An axial inclination of a tooth is a chief variant in anatomical morphology. While examining an axial inclination, only the crown surface is evaluated and it is assumed that the root follows the same alignment. But while examining an anterior tooth there can be discrepancy in longitudinal axes of the crown of a tooth and root [2]. The corresponding angle between these two longitudinal axes is defined as the crown to root angle and therefore, the CA is the supplementary angle of the crown to root angulation, used to correlate the angular difference between the two axes [3]. The CA of single rooted teeth is crucial in the field of dentistry especially orthodontics, prosthodontics, periodontics and implantology since a variation in CA might affect the prognosis of many dental treatments [4].

The CA has been investigated most often by using lateral cephalometric radiographs in different malocclusions. However, superimposition of structures is a main disadvantage in lateral cephalometric radiographs [3]. Presently, CBCT has been used because of its three dimensional capacity to evaluate the anatomic structures of the maxilla, mandible, and teeth [5]. CBCT is a revolutionary discovery that is being widely used in all fields of dentistry, including orthodontics, endodontics, oral surgery/pathology, periodontics, and implant treatment planning [4,6]. CBCT data provides additional benefits like availability of the images produced, and reducing the storage expenses [5]. Hence these advanced radiographic assessment methods provide more precise and reliable tools for obtaining tooth measurements.

However, research about the use of 3D CBCT to evaluate the crown-root morphology of maxillary central incisor remains rare. In addition, no related research exists regarding analysis and comparison of CA in Arabian population.

Therefore, this study aimed to determine and compare the mean CA for maxillary central incisor in different types of malocclusion using 3D CBCT records combined with computer aided measurement technology.

The objectives were as followed:Determine and compare the mean CA for maxillary central incisor based upon genderDetermine and compare the mean CA for maxillary central incisor in patients with different nationalities namely, Saudi, Jordanians and EgyptiansTest for significant differences in the CA of maxillary central incisor with different molar malocclusions.

## 2. Materials and Methods

A cross-sectional study of secondary data was conducted in a hospital-based setting. Ethical approval was granted by the local bioethics committee (approval no. LCBE 9-16-8/39).

Sample size estimation was done by using GPower software (version 3.0). Sample size was estimated for F test and ANOVA: Fixed effects, Omnibus one way and 4 groups with unequal sample size were chosen. The maximum sample size was calculated for difference in CA with respect to maxillary central incisor among different classes of molar malocclusion [3]. Minimum total sample size of 280 was found to be sufficient for an alpha of 0.05, power of 95%, 0.25 as effect size [3]. So, a total size was fixed at 400 for equal distribution of sample among four groups, viz. Class I, Class II div 1, Class II div 2 and Class III. 

### 2.1. Sample Characteristics

CBCT scans were reclaimed from the hospital records (from September 2019 to December 2021) and were acquired with SCANORA 3Dx (Tuusula, Finland) set at the following parameters: 90 kilovolts, 10 milliamperes, scanning time of 20 s with an extended field of view mode (100 × 100 mm^2^) and with resolution of 0.38 mm (voxel size of 0.25 mm). A total of 400 CBCT images were selected using computerized random technique. The inclusion criteria [3] considered were (i) age-group of 17–55 years; (ii) presence and complete eruption of all morphologically normal permanent teeth with exception of third molars; and (iii) High quality images with good density and contrast. Ethnicity was verified from the folder. The exclusion criteria [3] considered were (i) presence of dental prostheses; (ii) mixed dentition severe crowding or hypodontia in the anterior region; and (iii) worn incisal edges.

The scans were standardized by orienting the head in natural head position in three planar views. Firstly, the odontoid process of atlas bone in an axial view was traced. Following this, the head was aligned so that the odontoid process and the midline of maxilla would lie equally bisected by a vertical line. Secondly, for the sagittal section, the head was oriented by connecting the anterior nasal spine to the posterior nasal spine and this line was made parallel with the bottom on the monitor. Finally, the coronal section was aligned by approximating the mandibular condyles so that their shape and size are relatively equal. Then the head was rotated so that a vertical line bisects the midline of the oropharyngeal airway.

### 2.2. Studied Parameters

After characterizing each patient by their molar classification, the corresponding CBCT Digital Imaging and Communications in Medicine (DICOM) files were anonymized by converting all identifiable information into a random number. This number was recorded into an excel spreadsheet in which all other information pertinent to the patient were recorded. The selected records were used for the measurements of CA for maxillary central incisor using the linear and angular measurement tools provided in the dedicated software (Scanora 3D:OnDemand, Daejeon, Korea). The images were displayed on a TFT 27-inch monitor with 1280 × 1024 pixel screen resolution. The selected CBCT data were categorized into four groups according to Angle’s molar malocclusion classification- Class I, Class II div 1, Class II div 2, and Class III. The selected radiographs were independently examined by an oral radiologist and an orthodontist. All interpretations were done as per accepted standards, and any conflicts were decided by consensus.

### 2.3. Measurement of Crown to Root Angle (x)

The selected scans were visualized using 3D:OnDemand software. For the axial section of maxilla/mandible, the slices were set to have a thickness of 2.0 mm with slice increments set at 0.1 mm. The axial slice with the best view of the anterior teeth of maxilla/mandible was then chosen. The sagittal slice was created by moving the cursor to the distal and mesial side of the tooth so that the slice is centered on the midline of the tooth. Once the sagittal slice was directly centered on the longitudinal axis, the tooth can then be measured.

Three points were used to measure the crown to root angle (x). Incisor superioris that is an undamaged incisal edge was taken as the first point. A line connecting the facial and the lingual cementoenamel junction was considered as the second point and the third point was the apex of the root [3]. 

The CA was calculated by subtracting the crown to root angle from 180°. Therefore, the formula for the CA is 180-x. (Figure 1)

### 2.4. Measurement of Error

At two weeks interval, the readings were repeated to assess the systemic and random errors for randomly selected 40 CBCT images. Two-sample *t*-test was used for calculating systemic errors. Intra-class correlation of value > 0.91 was found that is adequate as per Stirrup (1993) [7]. All the measurement pairs showed a *p*-value > 0.1, suggesting that there was no systemic and random bias in these analyses as specified by Houston (1983) [8].

### 2.5. Statistical Analysis

Data was analyzed using Statistical Package for Social Sciences version 21.0 (IBM Corp, Armonk, NY, USA). The CA was presented as mean and standard deviation. Shapiro Wilk test was used to check the normality of the data and the data was found to be normal. Inferential statistics were performed using parametric tests of significance since the collected data was normal and continuous in nature. Inferential statistics were performed using one way Analysis of Variance (ANOVA) and repeated measures ANOVA test. One way ANOVA test was used to find out the significance of malocclusion class-wise and racial differences in the CA. Independent t test was used to check the significance of gender-wise differences. Post hoc pairwise comparison was done using post hoc Boneferroni’s test. The level of statistical significance was set at <0.05.

## 3. Results

The age ranged from 17 to 54 years, and the average age was 33 years.

Mean CA—The mean CA for the maxillary central incisor in Class I, Class II div 1, Class II div 2 and Class III sample was found to be 6.06 ± 1.75°, 6.21 ± 0.83°, 10.54 ± 1.85° and 5.42 ± 0.75° respectively. Thus the mean CA for Class II div 2 exhibited significantly higher crown-root variation as compared to other groups (*p* < 0.0001; Table 1). 

Gender-wise comparison—A total of 228 (57%) male sample and 172 (43%) female sample were included in the study. Males’ sample (6.18 ± 1.89° for Class I, 6.37 ± 0.84° for Class II div 1, 10.92 ± 1.99° for Class II div 2 and 5.69 ± 0.76° for Class III) showed greater value of CA for each group as compared to the females sample (5.79 ± 1.35° for Class I, 5.93 ± 0.75° for Class II div 1, 10.17 ± 1.66° for Class II div 2 and 5.10 ± 0.60° for Class III). This difference was statistically significant for all the groups other than for Class I (*p* < 0.05; Table 2). 

Race-wise comparison—The mean CA for Class I and Class III was found be higher for Jordanian subpopulation (6.29 *±* 1.33° and 5.60 *±* 0.63° respectively) whereas for Class II div 1 and 2, the Egyptian subpopulation showed higher value of mean CA (6.34 *±* 0.97° and 10.99 *±* 1.76° respectively). However, post hoc pairwise comparisons showed statistically insignificant findings (*p* < 0.05; Table 3).

Pairwise comparisons of different malocclusion groups—Significant difference was found on pairwise comparisons among different malocclusion groups other than for group Class I/Class II div 1 (Table 4).

## 4. Discussion

Collum angle plays an important role in dentistry especially in the anterior teeth that forms an esthetic zone [3]. In case of large CA, it becomes difficult to construct the core in patients needing post placement in teeth. Similarly, in patients with compromised periodontium- dehiscence, root prominence, and soft tissue esthetics may be affected [4]. The necessity to use an angled abutment when placing anterior implants may lead to stress concentration on the buccal side of the fixture, causing post-surgical tension in the gingiva that will eventually lead to recession and other unwarranted cosmetic defects [3]. Also, Heravi F et al. in their study had showed that retraction of maxillary central incisors in Class II div 2 patients resulted in forces that were 1.18x higher than in the Class I maxillary incisors, thereby supporting the fact that there is increase in stress generation in orthodontics with large CA in natural dentition. But on application of intrusive forces, the teeth with larger CAs had lower stress distribution to the periodontal ligament [9]. 

Previous research mainly based on the tracing lateral cephalograms for the measurement of CA [3]. Although lateral cephalometric templates are standardized, it is apparent that morphological variations, such as the CA, may not be justified for in the standardization process due to the superimposition of surrounding structures, magnifying distortion and unclear manual tracing of the tooth boundary that might affect the accuracy in measurements of CA [3]. The other previously used method employed the use of extracted teeth. However, the difficulty with extracted teeth is that they cannot be classified by molar classification and are difficult to obtain in large volumes [3]. With the advent of CBCT, superimposition issues with lateral cephalograms are overcome and clear three-dimensional imaging of tooth bone structure and precise measurement via digital software is possible [10]. Also, these records can precisely reproduce a patient dataset in a secondary environment, and allows the practitioner to create and manipulate these records as a 3D “virtual patient”. In this way, the use of CBCT technology provides a more efficient and practical method of measuring CA [11]. Furthermore, the application of CBCT in tooth morphometry remains rare [3]. 

Many theories are proposed regarding the development of larger CAs in Class II div 2 malocclusion that is inclusive of both hereditary and environmental factors [12]. According to Srinivasan B et al., the main cause for larger CAs were the lower lip pressure and its position on upper central incisor crown [13]. According to Lapatki B et al., the levels of the lip line and lip pressure are external factors of bend in long axes between the crown and root in upper anterior teeth [14]. Similarly, the reason for deviant CAs in Class II div 2 malocclusions is thought to be due to the lingually “bent” upper central incisors, thereby leading to the development of deep bites in these malocclusions [3]. 

This study assessed the CA of maxillary central incisor utilizing 3D CBCT images. The exclusivity of the present study lies upon the following statements: (1) 400 CBCT images; (2) assessment of CA by new method; and (3) first time in Arabian subpopulation using this novel method. In the current study, the mean CA for Class II div 2 exhibited significantly higher crown-root variation as compared to other groups. In general, this finding is in line with many previous studies [1,2,3,9,15,16,17,18,19,20]. But unlike the previous studies, Harris EF et al. found that Class III malocclusions had significantly higher mean CA as compared to Class I and II [21].

In the current study, the males sample showed significantly greater value of CA for all the groups as compared to the females. Although this finding is in accordance with the study done by Shailaja AM et al. but they have made overall comparison of mean CA between males and females whereas in the current study the comparison was made for each group [22].

In the present study, the mean CA for Class I and Class III was found be higher for Jordanian subpopulation whereas for Class II div 1 and 2, the Egyptian subpopulation showed higher value of mean CA. However, post hoc pairwise comparisons showed statistically insignificant findings among the racial groups. On contrary, one study had shown differences in ethnic and racial norms wherein Asian races have been noted to have larger CAs due to their ethnic propensity towards bimaxillary protrusion [16].

Differences observed in this study were statistically relevant when Class II div 2 cases were compared to other groups except for group Class I/Class II div 1. Similar findings were reported by most of the studies that individually compared Class II div 2 patients to specific malocclusions. Hence it is proven that upper central incisors are morphologically peculiar. 

Table 5 shows the various studies highlighting the measurements and comparison of Class II div 2 with other malocclusion groups [1,2,3,15,16,17,18,19,20,21,23,24].

## 5. Limitations

Presence of artifacts on CBCT scans was the main limitation of this study. Although most of the scans that had poor radiographic quality were excluded, there were few images where noise posed some issues and caused hindrance in clear view of the upper central incisor root apex. Secondly, the sample size for the Class II div 2 malocclusions was comparatively small. Thirdly, the Class III malocclusions were not further stratified into different groups that would have helped in distinguishing if the CAs are affected by the deflection of the mandibular arcade.

## 6. Conclusions

Among patients with different types of malocclusion, Class II div 2 malocclusion group showed a significantly greater CA. Males sample showed greater value of CA for each group as compared to the females and this difference was statistically significant for all the groups other than for Class I. The post hoc pairwise comparisons between the races showed statistically insignificant findings. Significant difference was found on pairwise comparisons among different malocclusion groups other than for group Class I/Class II div 1. Thus, according to the results of the current study, it is recommended that during treatment planning of dental procedures, either CA or crown-root angle should be evaluated especially using CBCT technique. The data and results of this study can be used for future researches.

## Figures and Tables

**Figure 1 diagnostics-12-01506-f001:**
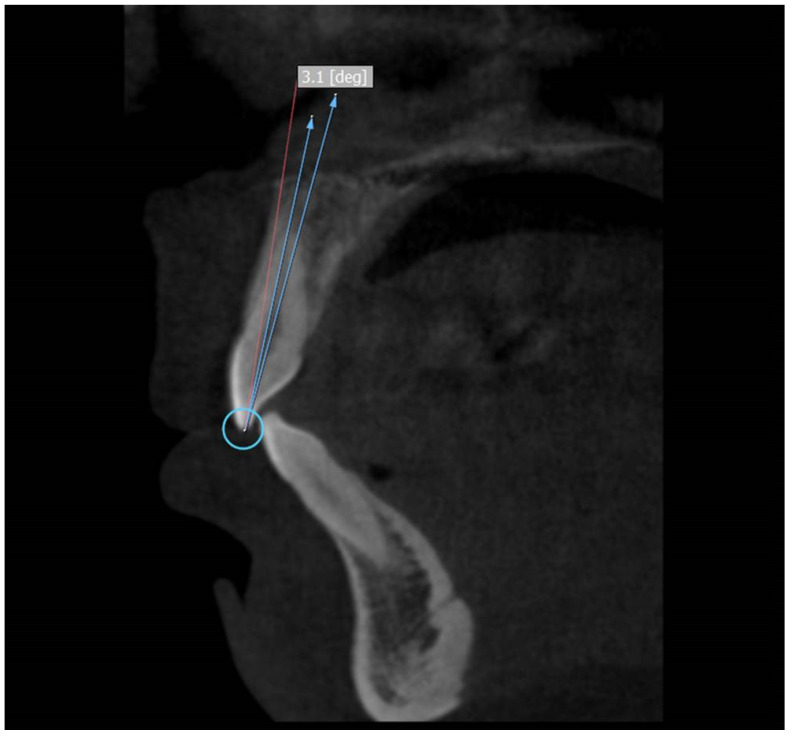
Use of the 3D:OnDemand software to locate the specific points of median sagittal sections for maxillary anterior teeth and to measure crown to root angle.

**Table 1 diagnostics-12-01506-t001:** Mean CA of maxillary central incisor in different malocclusion types (shown in degree).

Malocclusion Type	N	Mean	SD	95% Confidence Interval		F Value	*p* Value (One Way ANOVA Test)
				Lower Bound	Upper Bound		
Class I	100	6.06	1.75	5.71	6.40	283.9	<0.0001 *
Class II div 1	100	6.21	0.83	6.04	6.37		
Class II div 2	100	10.54	1.85	10.17	10.90		
Class III	100	5.42	0.75	5.27	5.56		
Total	400	7.05	2.46	6.81	7.30		

N—Sample number; *—Significant.

**Table 2 diagnostics-12-01506-t002:** Gender-wise comparison of CA.

Variables	Gender	N	Mean	SD	t Value	*p* Value (Independent *t* Test)
Class I	M	69	6.18	1.89	1.029	0.306
	F	31	5.79	1.35		
Class II div 1	M	62	6.37	0.84	2.643	0.01 *
	F	38	5.93	0.75		
Class II div 2	M	44	10.92	1.99	2.258	0.026 *
	F	56	10.17	1.66		
Class III	M	53	5.69	0.76	4.202	<0.0001 *
	F	47	5.10	0.60		

* Significant.

**Table 3 diagnostics-12-01506-t003:** Racial-wise comparison of CA.

Variables	Nationality	N	Mean	SD	F Value	*p* Value (One Way ANOVA Test)
Class I	Saudi	52	5.90	2.04	0.444	0.643
	Jordan	23	6.29	1.33		
	Egypt	25	6.16	1.45		
Class II div 1	Saudi	51	6.13	0.79	0.605	0.548
	Jordan	19	6.20	0.71		
	Egypt	30	6.34	0.97		
Class II div 2	Saudi	55	10.37	1.92	0.965	0.385
	Jordan	21	10.43	1.77		
	Egypt	24	10.99	1.76		
Class III	Saudi	49	5.41	0.80	1.271	0.285
	Jordan	23	5.60	0.63		
	Egypt	28	5.27	0.75		

**Table 4 diagnostics-12-01506-t004:** Post hoc pairwise comparisons using Boneferroni test.

Malocclusion Groups Compared	*p* Value
Class I/Class II div 1	0.872
Class I/Class II div 2	<0.0001 *
Class I/Class III	0.007 *
Class II div 1/Class II div 2	<0.0001 *
Class II div 1/Class III	<0.0001 *
Class II div 2/Class III	<0.0001 *

* Significant.

**Table 5 diagnostics-12-01506-t005:** Various studies highlighting the measurements and comparison of Class II div 2 with other malocclusion groups.

Authors’	N	Class I	Class II Div		Class III	Measurements	Comments
			1	2			
Delivanis, H.; Kuftinec, M. (1980)	106	17	27	53	9	Collum angle	Class I, Class II div 1, and Class III data pooled for comparison with “severe” Class II div 2 cases only
Williams, A.; Woodhouse, C. (1983)	191	65	66	29	31	Crown/root angle	Cephalograms selected having “clarity of upper central incisors”
Bryant, R. et al. (1984)	100	25	25	25	25	Crown/root angle	Only included “severe” cases Priority given to images with “optimal sharpness and clarity”
Harris, E.F. et al. (1993)	79	24	34	0	21	Collum angle	Class II div 2 cases excluded
Korda, R.A. et al. (2000)	43	0	16	27	0	Ratio of root length/crown length and root/crown angle	Cephalograms recorded using two different cephalostats Compared only Class II div 1 with Class II div 2
McIntyre, G.T.; Millett, D.T. (2003)	361	71	139	60	91	Crown-root shape	Monitor-displayed digital cephalograms
Shen, Y.W. et al. (2012)	124	33	32	28	31	Collum angle	Lateral cephalometric radiographs
Li, M. et al. (2016)	36	18	0	18	0	Crown-root angle and surface-shaft angle	CBCT combined with computer aided measurement technology
Israr, J. et al. (2016)	60	0	30	30	0	Collum angle	Lateral cephalometric radiographs
Feres, M.F.N. et al. (2018)	48	16	16	16	0	Collum angle	Used CBCT images
Wang, X.M. et al. (2019)	66	24	20		22	Collum angles and labial surface angles with different skeletal malocclusions	Used CBCT images No subdivision of Class II malocclusion
Elangovan, B. et al. (2020)	60	15	15	15	15	Collum angle	Used CBCT images Studied maxillary and mandibular anterior teeth
Current study (2022)	400	100	100	100	100	Collum angle (mean), gender-wise and race-wise comparison	3D CBCT

## Data Availability

The data set used in the current study will be made available on request from Rakhi Issrani; dr.rakhi.issrani@jodent.org.

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
