# Peer review of "Cone Beam Computerized Tomographic Analysis of Collum Angle of the Maxillary Central Incisors in Different Types of Malocclusion: Comparative Assessment in Saudi, Jordan and Egypt Subpopulation"

_diagnostics, 2022, doi:10.3390/diagnostics12061506_

Round 1

Reviewer 1 Report

Dear Authors, thank you for submitting your paper.

The aim of the present study was to determine and compare the mean CA for maxillary central incisor in different 67 types of malocclusion using 3D CBCT records combined with computer aided measurement technology.

I congratulate the authors for this very relevant research, which will add to the dental field.

The study is novel and the first one in literature for the topic. It appears well structured, correctly carried out and written without logical or factual errors

Methodological aspects are deeply cleared in the manuscript. The authors reported an Interesting investigations  of collum angle of the maxillary central incisors in different types of malocclusion

The topic is in line with the journal aim.

-Please introduce in the sample size section the data used for the calculation of the sample size.

-Data reported in the Methods section are appropriate and precisely described;.

-Results are reported clearly and adequately supported by Tables.

-I suggest to the Authors to improve their reference list adding the following recent articles about the use of CBCT in dentistry:

https://doi.org/10.1007/s00330-020-06905-7

DOI:10.1186/s13005-022-00308-3

https://doi.org/10.1186/s40510-021-00380-6

The Conclusions are correctly stated and supported by the findings obtained from the present study.

According to this Reviewer’s consideration, novelty and quality of the paper, publication of the present manuscript is recommended.

Author Response

Dear Reviewer,

Thank you for your valuable feedback.

Warm regards

Authors’ reply to reviewer’s comment

Reviewer’s comment

Authors’ reply

Page no.

Methodological aspects are deeply cleared in the manuscript. The authors reported an interesting investigations  of collum angle of the maxillary central incisors in different types of malocclusion

The topic is in line with the journal aim.

Thank you for your valuable feedback & appreciation.

-Please introduce in the sample size section the data used for the calculation of the sample size.

Added

Page 2

-Data reported in the Methods section are appropriate and precisely described;.

-Results are reported clearly and adequately supported by Tables.

Thank you for your valuable feedback & appreciation.

-I suggest to the Authors to improve their reference list adding the following recent articles about the use of CBCT in dentistry:

https://doi.org/10.1007/s00330-020-06905-7

DOI:10.1186/s13005-022-00308-3

https://doi.org/10.1186/s40510-021-00380-6

Added

Page 2

The Conclusions are correctly stated and supported by the findings obtained from the present study.

Thank you for your valuable feedback & appreciation.

Reviewer 2 Report

1) First of all, this article is quite similar to the previous studies in the literature such as :

- Elangovan B et al, Int Orthod 2020;18:468-479 (CBCT study)

- Wang XM et al, Prog Orthod 2019;20:20 (CBCT study)

- Feres MFN et ak, J Orthod Sci 2018;7:6 (CBCT study)

Thus, this article shows very low novelty and does not add much information to the literature. Moreover, the authors did not cite Elangovan B et al and Wang XM et al. This indicates the lack of literature review.

2) p2, l1-2; Reference #1 is about the maxillary central incisor only, which is not appropriate to support the generalized statement.

3) Reference #2; The reference is thesis, which has not been subject to peer review.

4) Introduction, 1st Para; How does one can define the “axis of the crown” and “axis of the root”? Unclear.

5) p2, l9, “crucial”; How crucial? It looks like prognosis of the artificial restoration such as prosthodontic and dental implant treatment may be affected by the variation in the CA. However, is there any evidence that the variation in the CA of the “natural” teeth affect the prognosis of the orthodontic treatment?

6) Reference #4; The reference cannot be found in PubMed. Replace with more popular article with more availability.

7) Introduction, 13rd Para; The statement is inappropriate, because there are several articles already available on this issue.

8) p4, Figure 1; Don’t understand the explanations in the figure.

9) Discussion, 1st Para; There are a lot of argument that is irrelevant to the CA of the natural teeth.

10) Reference #9; The reference is also thesis, which has not been subject to peer review.

Author Response

Dear Reviewer,

Thank you for your valuable feedback.

Warm regards

Authors’ reply to reviewers’ comments

Reviewer’s comment

Authors’ reply

Page no.

1) First of all, this article is quite similar to the previous studies in the literature such as :

- Elangovan B et al, Int Orthod 2020;18:468-479 (CBCT study)

- Wang XM et al, Prog Orthod 2019;20:20 (CBCT study)

- Feres MFN et ak, J Orthod Sci 2018;7:6 (CBCT study)

Thus, this article shows very low novelty and does not add much information to the literature. Moreover, the authors did not cite Elangovan B et al and Wang XM et al. This indicates the lack of literature review.

These articles as recommended by the reviewer are added.

Page 8.

2) p2, l1-2; Reference #1 is about the maxillary central incisor only, which is not appropriate to support the generalized statement.

Another article that supports this statement is added.

Page 2

3) Reference #2; The reference is thesis, which has not been subject to peer review.

Deleted.

--

 4) Introduction, 1st Para; How does one can define the “axis of the crown” and “axis of the root”? Unclear.

By axis, we mean the inclination of crown and root. The inclination of crown is visible clinically but the axis of root (that sometimes doesn’t follow the inclination of crown) can be studied by radiographs.

5) p2, l9, “crucial”; How crucial? It looks like prognosis of the artificial restoration such as prosthodontic and dental implant treatment may be affected by the variation in the CA. However, is there any evidence that the variation in the CA of the “natural” teeth affect the prognosis of the orthodontic treatment?

Evidence that the variation in the CA of the “natural” teeth affect the prognosis of the orthodontic treatment is mentioned in the paper.

Page 7

6) Reference #4; The reference cannot be found in PubMed. Replace with more popular article with more availability.

Few recent articles are added.

Page 10-11

7) Introduction, 13rd Para; The statement is inappropriate, because there are several articles already available on this issue.

We completely agree with the reviewer that there are several papers on the same topic but no related research exists regarding analysis and comparison of CA in Arabian population.

--

8) p4, Figure 1; Don’t understand the explanations in the figure.

The figure caption is elaborated.

Page 4

9) Discussion, 1st Para; There are a lot of argument that is irrelevant to the CA of the natural teeth.

This part is edited.

Page 6

10) Reference #9; The reference is also thesis, which has not been subject to peer review.

Deleted.

--

Reviewer 3 Report

Methodological Biases exist

(The Authors must see my remarks)

Author Response

Dear Reviewer,

Thank you for your valuable feedback.

The valuable comments in the pdf file as provided by you are edited and highlighted in yellow color in the main manuscript.

Warm regards

Authors’ reply to reviewer’s comment

Reviewer’s comment

Authors’ reply

Page no.

Methodological Biases exist

The authors’ have tried to their best to reduce the bias as much as possible.

The selected radiographs were independently examined by an oral radiologist and an orthodontist. All interpretations were done as per accepted standards, and any conflicts were decided by consensus.

Page 3

Reviewer 4 Report

The manuscript is of interest. There are however, some methodological issues requiring authors' attention:

  • Who performed the measurements? Were examiners trained and calibrated prior to study beginning? Please report concordance between examiners.
  • How was repeatability assured in performing the measurements? 
  • In figure 1, it is shown a sagittal image. How was volume slicing managed?
  • How were potential errors in patient positioning managed?

Author Response

Dear Reviewer,

Thank you for your valuable feedback.

Warm regards

Authors’ reply to reviewer’s comment

Reviewer’s comment

Authors’ reply

Page no.

Who performed the measurements? Were examiners trained and calibrated prior to study beginning? Please report concordance between examiners.

The selected radiographs were independently examined by an oral radiologist and an orthodontist. All interpretations were done as per accepted standards, and any conflicts were decided by consensus.

Page 3

How was repeatability assured in performing the measurements?

“Measurement of error” is added the methodology section.

Page 4

In figure 1, it is shown a sagittal image. How was volume slicing managed?

Yes, it is a sagittal image.

--

How were potential errors in patient positioning managed?

The positional errors were tried to be as less as possible by using the same CBCT machine and all the images were taken by a single, trained and experienced oral radiologist.

--

Round 2

Reviewer 2 Report

Unfortunately, the authors failed to sufficiently respond to my comments, especially in novelty. The authors themselves stated “We completely agree with the reviewer that there are several papers on the same topic but no related research exists regarding analysis and comparison of CA in Arabian population.” This is what I pointed out as a lack of novelty. This study is a mere confirmation of a known fact with different races.

Author Response

Dear Reviewer,

Once again thank you for your valuable feedback.

Warm regards,

Dr. Rakhi Issrani

Authors’ reply to reviewers’ comments

Reviewer’s comment

Authors’ reply

Page no.

Unfortunately, the authors failed to sufficiently respond to my comments, especially in novelty. The authors themselves stated “We completely agree with the reviewer that there are several papers on the same topic but no related research exists regarding analysis and comparison of CA in Arabian population.” This is what I pointed out as a lack of novelty. This study is a mere confirmation of a known fact with different races.

Thank you for your valuable feedback.

Though the method is not novel, but this study has been done for the first time in Arabian subpopulation and compared among 3 races.

--

Reviewer 3 Report

Few corrections are required

(The Authors must see my remarks)

Author Response

Dear Reviewer,

Once again thank you for your valuable feedback.

Warm regards,

Dr. Rakhi Issrani

Reviewer’s comment

Authors’ reply

Page no.

Few comments in the pdf.

The references are added as mentioned in the pdf file by the respected reviewer.

Page 2 & 3

Reviewer 4 Report

Please specify how volume slicing was standardized in the CBCT datasets, and add the thickness of the slices for each examination.

Author Response

Dear Reviewer,

Once again thank you for your valuable feedback.

Warm regards,

Dr. Rakhi Issrani

Reviewer’s comment

Authors’ reply

Page no.

Please specify how volume slicing was standardized in the CBCT datasets, and add the thickness of the slices for each examination.

Included in the methodology section.

Page 7